# Association between Intake of Total Dairy and Individual Dairy Foods and Markers of Folate, Vitamin B_6_ and Vitamin B_12_ Status in the U.S. Population

**DOI:** 10.3390/nu14122441

**Published:** 2022-06-13

**Authors:** Christopher J. Cifelli, Sanjiv Agarwal, Victor L. Fulgoni III

**Affiliations:** 1National Dairy Council, 10255 West Higgins Road, Suite 900, Rosemont, IL 60018, USA; 2NutriScience LLC, East Norriton, PA 19403, USA; agarwal47@yahoo.com; 3Nutrition Impact LLC, Battle Creek, MI 49014, USA; vic3rd@aol.com

**Keywords:** NHANES, milk, yogurt, cheese, biomarkers, pyridoxal 5′-phosphate, cobalamin

## Abstract

Vitamin B_6_, B_12_ and folate are required for energy metabolism and have been identified as nutrients of concern for certain population groups. This study examined the cross-sectional association between the consumption of dairy (total dairy, milk, yogurt and cheese) and biomarkers and adequacy for these nutrients in a nationally representative sample. Twenty-four-hour dietary recall data and concentrations of RBC folate (ng/mL), serum folate (ng/mL), and serum vitamins B_6_ (nmol/L) and B_12_ (pg/mL) were obtained from the National Health and Nutrition Examination Survey 2001–2018 (*n* = 72,831) and were analyzed by linear and logistic regression after adjusting for demographic variables. Significance was set at *p* < 0.01. Mean intakes of total dairy were 2.21, 2.17, 1.83 and 1.51 cups eq among consumers aged 2–8, 9–18, 19–50 and 51+ years, respectively. Higher intakes of total dairy as well as individual dairy foods (especially milk and yogurt) were positively associated with serum and RBC folate, serum vitamin B_6_ and serum B_12,_ and generally, with 9–57% lower risk of inadequate or deficient levels of these vitamins. These findings suggest that encouraging dairy consumption may be an effective strategy for improving micronutrient status and provide continued evidence to support the current dietary recommendations for dairy and dairy products.

## 1. Introduction

Vitamin B_6_, B_12_ and folate are coenzymes/cofactors, which participate in one carbon metabolism and mediate numerous metabolic and cellular functions. They help in the conversion of homocysteine into methionine and their low intakes have been associated with elevated plasma homocysteine levels, an independent risk for coronary heart disease and stroke, and correlated with decreased performance on cognitive tests [1,2,3,4]. Vitamin B_6_ (pyridoxal 5′-phosphate and pyridoxamine 5′-phosphate) is involved in more than 140 enzyme reactions affecting the metabolism of amino acids, lipids and neurotransmitters [5]. Vitamin B_12_ (cobalamin) is a cofactor for the remethylation of homocysteine and isomerization of l-methylmalonyl-CoA to succinyl-CoA and is indispensable for proper RBC formation, normal neurological function and DNA synthesis [6]. Folate is an essential component of key metabolic processes, including methylation, cell proliferation and DNA replication [7], and may have cognitive benefits during aging [8].

Vitamin B_12_ has been identified as a nutrient of concern for some older adults because the ability to absorb this nutrient can decrease with age and use of certain medications can decrease absorption [9]. Indeed, the probability of vitamin B_6_ and B_12_ deficiencies are higher in older population groups [10]. Vitamin B_12_ deficiency affects cell division and may lead to megaloblastic anemia and neuropathy [11]. Folate/folic acid is identified as a nutrient of concern among premenopausal women in the first trimester of pregnancy due to the high prevalence of inadequate folic acid intakes observed in women who are or are capable of becoming pregnant [12]. Inadequate intakes of folate among women are associated with increased risk of giving birth to infants with neural tube defects (NTDs) [1]. However, in 1998, all enriched grain products were required to be fortified with folic acid by US FDA, and this mandatory folic acid fortification resulted in a 35% decreased birth prevalence of NTDs in the United States [13]. Beyond the U.S., folate and vitamin B12 deficiencies appear to be a public health problem in a majority of countries for which national surveys were available [14].

Milk and dairy products are key sources of macro- and micronutrients and contribute to overall diet quality in both children and adults [15,16,17,18]. Accordingly, dairy products are an integral part of national dietary guidelines [9]. However, about 90 percent of the U.S. population does not meet dairy intake recommendations [9], and milk and dairy product intake has steadily declined from 2.21 cups eq in 2003–2004 to 1.70 cups eq in 2017–2018 among adolescents but did not change (1.57 cups eq in 2003–2004 and 1.45 cups eq in 2017–2018) among adults [19]. Per-capita daily consumption of fluid milk, which is a good source of vitamin B_12_, in the US, has decreased by over 40%, from 0.57 cups eq in 2003–2004 to 0.33 cups eq in 2017–2018 [20]. While the decrease in dairy intake is due to several factors, there has been a strong push recently by some to shift to a plant-based diet and to limit or eliminate animal-based foods for sustainability reasons [21,22,23,24]. This could lead to a decreased intake of certain micronutrients that are predominantly found in animal-based foods, such as vitamin B_12_.

Given the inextirpable link between vitamin B_6_, B_12_ and folate, we were interested in better understanding the relationship between total dairy food intake and milk, cheese and yogurt separately on nutrient status using accepted biomarkers of these nutrients. We hypothesize that higher intakes of dairy foods will be associated with improved status of these B vitamins.

## 2. Materials and Methods

### 2.1. Database

Data from What We Eat In America (WWEIA), the dietary intake component of the National Health and Nutrition Examination Survey (NHANES) for nine separate cycles from 2001 to 2018) were combined to access dairy intake and its association with certain nutrient biomarkers. NHANES is a now continuous series of cross-sectional surveys that use a complex, stratified multistage probability sample of a nationally representative, civilian noninstitutionalized US population identified from specified household clusters. A detailed description of NHANES procedures including the subject recruitment, survey design, and data collection is available online [25] and all data obtained for this study are publicly available at: http://www.cdc.gov/nchs/nhanes/ (accessed on 26 April 2021). NHANES protocols were approved by the National Center for Health Statistics (NCHS) Ethics Review Board and all participants or proxies (i.e., parents or guardians) provided written informed consent. The present study was a secondary data analysis which lacked personal identifiers and, therefore, was exempt from additional approvals by an Institutional Review Board.

### 2.2. Participants

Data from children and adults aged 2+ years (*n* = 84,629) participating in nine NHANES cycles (2001–2002, 2003–2004, 2005–2006, 2007–2008, 2009–2010, 2011–2012, 2013–2014, 2015–2016 and 2017–2018) were extracted for the current analysis. However, those with unreliable data, incomplete day-1 dietary recalls as determined by the USDA (*n* = 10,163), pregnant or lactating females (*n* = 1631) and with no day-1 dietary intake (*n* = 4) were excluded, and the final sample size after exclusions was 72,831 children and adults.

### 2.3. Dietary Intakes

Dietary intakes were estimated using the day-1 24 h dietary recall in-person interview that included the amount and a description of the individual foods and beverages consumed on the previous day (midnight to midnight). Complete descriptions of the dietary interview methods for NHANES are provided elsewhere [25]. Intakes of dairy components (milk, yogurt and cheese) were assessed by using the NHANES-cycle-specific MyPyramid Servings database (MPED) and Food Patterns Equivalent Database (FPED) variables as cup equivalents/day [26,27]. Milk included whole, reduced fat, low fat and non-fat milk, flavored milk and milk used in other foods/mixed dishes (e.g., mashed potatoes). Yogurt included both plain, flavored, Greek and regular, and all fat varieties of yogurt. Cheese included all varieties of regular, cottage and ricotta cheese and cheese used in other foods/mixed dishes (e.g., pizza, nachos, etc.). Consumers were defined as those consuming any amount of dairy components during the first (in-person) 24 h recall, while non-consumers were defined as subjects not consuming any dairy components during the first (in-person) 24 h recall.

### 2.4. Nutrient Biomarkers

As part of the NHANES in-person health examination in the Mobile Examination Center, participants provided a blood specimen for laboratory analyses. Values for serum folate, RBC folate, serum vitamin B6 (pyridoxal 5′-phosphate), and serum vitamin B12 (serum cobalamin) were obtained from the NHANES laboratory files [25]. Folate (serum and RBC), serum vitamin B_6_ and serum vitamin B_12_ were measured by microbiological assay, HPLC and electrochemiluminescence immunoassay, respectively. Detailed specimen collection and processing instructions, as well as quality control measures used, are provided in the NHANES Laboratory/Medical Technologists Procedures Manual [25]. Laboratory data were available only for the following NHANES cycles because not all nutrient biomarkers were measured in each NHANES cycle: serum and RBC folate—NHANES 2001–2018; serum vitamin B_6_—NHANES 2003–2010; and serum vitamin B_12_—NHANES 2001–2006 and 2011–2014. The biomarker cut-off levels used to identify inadequacy/deficiency were: inadequate RBC (erythrocyte) folate (<140 ng/mL); inadequate serum folate (<3 ng/mL); inadequate serum vitamin B_6_ (pyridoxal-5′-phosphate levels < 20 nmol/L); and serum vitamin B_12_ deficient (serum cobalamin < 200 pg/mL) [10].

### 2.5. Statistics

All analyses were performed using SAS 9.4 (SAS Institute, Cary, NC, USA) software after adjusting the data for the complex sampling design of NHANES, using appropriate survey weights, strata and primary sampling units. Separate analyses were conducted for the ages 2–8, 9–18, 19–50 and 51+ years. Consumer intake quartiles were calculated within each age group for each dairy type. Least-square means (and the standard errors) for each quartile of dairy intake and linear trends across consumer quartiles were determined for laboratory values of serum folate, RBC folate, serum vitamin B_6_ (pyridoxal 5′-phosphate) and serum vitamin B_12_ (serum cobalamin) using PROC SURVEYREG of SAS 9.4 after adjusting for age, gender and ethnicity. Regression coefficients (β) indicating the change in each nutrient status marker per-cup equivalent of dairy variable were determined. Logistic regression analysis was used to assess odds ratios (OR) and 99% confidence limits (lower confidence limit (LCL); upper confidence limit (UCL)) of meeting cutoff levels across quartiles of dairy intake with non-consumers as reference group after adjusting for age, gender and ethnicity. Results are not indicated if there were less than 5 subjects in any population group. Given the very large sample size a more conservative *p*-value of *p* < 0.01 was deemed significant.

## 3. Results

Table 1 shows the mean and quartiles of intake of total dairy, milk, yogurt and cheese by age group among consumers. Mean intake of total dairy and milk were higher for children aged 2–8 and 9–18 years than for adults 19–50 and 51+ years.

Table 2 shows the RBC folate status among non-consumers and across increasing consumer quartiles of total dairy, milk, yogurt and cheese. For children 2–8 years, RBC folate was inversely associated with increasing intake of milk (β, ng/mL RBC per cup eq. = −12.5 ± 2.4, P_linear trend_ < 0.01), and positively associated with increasing intake of cheese (β = 14.8 ± 4.9, P_linear trend_ < 0.01). RBC folate was also positively associated with increasing intake of yogurt (β = 59.2 ± 16.7, P_linear trend_ < 0.01) in adolescents 9–18 years. For adults 19–50 and 51+ years, respectively, RBC folate was positively associated with increasing intakes of total dairy (β = 5.02 ± 1.15 and β = 9.36 ± 2.07, P_linear trend_ < 0.01 for both), yogurt (β = 29.8 ± 8.4 and β = 64.8 ± 15.5, Plinear trend < 0.01 for both) and cheese (β = 9.45 ± 1.65 and β = 14.5 ± 3.7, P_linear trend_ < 0.01).

Table 3 shows the serum folate status among non-consumers and across increasing consumer quartiles of total dairy, milk, yogurt and cheese. For children 2–8 years, serum folate was inversely associated with increasing intake of total dairy (β, ng/mL per cup eq. = −0.56 ± 0.13, P_linear trend_ < 0.01) and milk (β = −0.61 ± 0.17, P_linear trend_ < 0.01). For adolescents 9–18 years and adults 19–50 and 51+ years, respectively, serum folate was positively associated with increasing intakes of total dairy (β = 0.31 ± 0.07, β = 0.31 ± 0.09 and β = 0.40 ± 0.12, P_linear trend_ < 0.01 for all) and milk (β = 0.53 ± 0.10, β = 0.54 ± 0.17 and β = 0.85 ± 0.17, P_linear trend_ < 0.01 for all). Serum folate was also positively associated with increasing intake of yogurt for those 9–18 and 19–50 years, respectively (β = 2.91 ± 0.85 and β = 3.68 ± 1.31, P_linear trend_ < 0.01 for both), and inversely associated with increasing intake of cheese (β = −0.47 ± 0.18, P_linear trend_ < 0.01) for those 51+ years.

Table 4 shows the serum vitamin B_6_ (pyridoxal 5′-phosphate) status among non-consumers and across increasing consumer quartiles of total dairy, milk, yogurt and cheese. Serum vitamin B6 was positively associated with increasing intake of total dairy and milk for those 2–8 years (β, nmol/L per cup eq. = 2.85 ± 0.97, P_linear trend_ < 0.01 for milk only); those 9–18 years (β = 1.94 ± 0.56, β = −2.96 ± 0.81, P_linear trend_ < 0.01 for both); and those 51+ years (β = 4.75 ± 1.47, β = 5.68 ± 2.03, P_linear trend_ < 0.01 for both). Increasing intakes of yogurt were also positively associated with serum vitamin B6 for those 19–50 years (β = 30.4 ± 8.1, P_linear trend_ < 0.01) and those 51+ years (β = 44.5 ± 8.7, P_linear trend_ < 0.01). Intake of cheese was not associated with serum vitamin B_6_ for any age group.

Table 5 shows the serum vitamin B_12_ (cobalamin) status among non-consumers and across increasing consumer quartiles of total dairy, milk, yogurt and cheese. Serum vitamin B_12_ was positively associated with increasing intake of total dairy and milk for those 2–8 years (β, pg/mL per cup eq. = 31.1 ± 4.8, β = 43.5 ± 6.0, P_linear trend_ < 0.01 for both); those 9–18 years (β = 18.6 ± 2.4, β = 26.7 ± 3.0, P_linear trend_ < 0.01 for both); and those 19–50 years (β = 12.3 ± 3.4, β = 20.7 ± 4.1, P_linear trend_ < 0.01 for both). Milk intake was also positively associated with serum vitamin B_12_ status for adults 51+ years (β = 15.6 ± 5.6, P_linear trend_ < 0.01). Intake of yogurt or cheese was not associated with serum vitamin B_12_ for any age group.

Table 6 shows the linear trend in odds ratios for not meeting the recommended levels of RBC folate (140 ng/mL), serum folate (3 ng/mL), serum vitamin B_6_ (20 nmol/L) and serum vitamin B_12_ (200 pg/mL) across consumer quartiles of dairy, milk, yogurt and cheese intake by age groups. Increasing quartiles of total dairy intake were associated with lower risk of having inadequate RBC folate (27%, 17% and 27% lower risk among those 9–18, 19–50 and 51+ years, respectively); inadequate serum folate (57%, 22% and 31% lower risk among those 9–18, 19–50 and 51+ years, respectively); inadequate serum vitamin B_6_ (11% and 16% lower risk among those 19–50 and 51+ years, respectively); and deficient in serum vitamin B_12_ (51% and 25% lower risk among those 9–18 and 51+ years, respectively). Increasing intake of milk intake was also associated with lower risk of having inadequate RBC folate (21% and 22% lower risk among those 9–18 and 51+ years, respectively); inadequate serum folate (46% and 26% lower risk among those 9–18 and 51+ years, respectively); inadequate serum vitamin B_6_ (9% and 14% lower risk among those 19–50 and 51+ years, respectively); and deficiency in serum vitamin B_12_ (40% and 23% lower risk among those 9–18 and 51+ years, respectively). Increasing intake of yogurt was also associated with lower risk of having inadequate serum folate (43% lower risk among those 19–50 years); and inadequate serum vitamin B_6_ (23% and 22% lower risk among those 19–50 and 51+ years, respectively). However, cheese intake was only associated with lower risk of having inadequate RBC folate (15% and 21% lower risk among those 19–50 and 51+ years, respectively); and inadequate serum folate (37% lower risk among those 9–18 years) but not with serum vitamins B_6_ or B_12_.

## 4. Discussion

The current cross-sectional analysis of data from nine cycles of NHANES showed increasing the consumption of total dairy as well as individual dairy foods (especially milk and yogurt) was associated with higher status of serum and RBC folate, serum vitamin B_6_ and serum B_12_, and generally, with lower risk of inadequacy/deficiency of these vitamins based on the levels of their biomarkers. Serum folate is indicative of recent intake of folate while RBC folate is indicative of body folate stores and long-term intake status; serum pyridoxal 5′-phosphate is generally viewed as the single best indicator of vitamin B_6_ status and serum cobalamin is an indicator of vitamin B_12_ status [10]. Intakes of dairy were also below the recommended amounts for all ages, suggesting that nutrition education that helps facilitate adherence to the Dietary Guidelines for Americans might increase dairy consumption and, in turn, contribute to higher nutrient status.

The results presented indicate that dairy foods or other foods (e.g., fortified cereals) consumed with dairy foods make significant contributions to the micronutrient requirements of children and adults, with the exception of those 2–8 years, where an inverse association of dairy (total and milk) with serum or RBC folate were observed and the reasons for this are not immediately apparent though may be related to quantity of consumption, as younger children consume less than older children and adults. Similar observations between intakes of dairy and micronutrients status were also reported earlier in diverse cohorts. Higher intakes of dairy were associated with improved blood concentrations of biomarkers for vitamin B_12_, folate, vitamin B_2_ and B_6_ in older Irish adults [28]. In an older Dutch population, higher intakes of dairy, meat, and fish and shellfish were significantly associated with higher serum vitamin B_12,_ with milk showing the most significant benefit [29]. Milk, fortified cereals and supplements were associated with higher plasma vitamin B_12_ concentrations in the Framingham Offspring study [30]. Plasma vitamin B_12_ status was associated with higher intakes of vitamin B_12_ from dairy products in the Norwegian Hordaland Homocysteine Study [31]. Milk and dairy were major contributors of these B vitamins in Greek school children [32], Spanish children and adults [33], Swedish 18–30-year-old adults [34] and in the Polish population [35]. Intake of dairy products was linked to higher intakes of nutrients, including folate, vitamin B_6_ and vitamin B_12,_ in participants of the Bogalusa Heart Study [36]. In a multiethnic population of Head Start Mothers, high milk/low sweetened beverage intakes were associated with higher mean intakes of numerous vitamins and minerals, including folate and B_6_ [37]. Dairy and alternatives were found to be the highest source of all one-carbon metabolism nutrients (methionine, folate, vitamins B_2_, B_6_ and B_12_) in Australian preschool children [38]. In a recent review, Obeid et al. [39] concluded that dairy consumption is a stronger determinant of vitamin B_12_ status than other animal products. Meeting dairy recommendations was also found to reduce inadequacies for all micronutrients, including a 20% decrease in folate inadequacy in a recent observational study with preschool children in the Philippines [40].

Increasing consumption of dairy as well as dairy components, especially milk and yogurt, was associated with increased B vitamin status and decreased risk of their deficiency. It is interesting to note that the per-cup linear increases across dairy intake quartile (β) were generally small compared to the respective cutoffs for inadequacy/deficiency, except for yogurt, where the relative increases were higher. For example, for RBC folate, the increases compared to inadequacy cutoff were 3.6% (age 19–50 years) and 6.7% (age 51+ years) for total dairy, and 6.7% (age 19–50 years) and 10.4% (age 51+ years) for cheese; for yogurt, the increases were 21.3% (age 19–50 years) and 46.3% (age 51+ years). Similarly, for serum folate, the increases compared to inadequacy cutoff were 10.3% (age 9–18 and 19–50 years) and 13.3% (age 51+ years) for total dairy, and 17.7% (age 9–18 and 19–50 years) and 28.3% (age 51+ years); for yogurt, the increases were 97% (age 9–18 years) and 122% (age 51+ years). Similar results were also found for vitamins B_6_ and B_12_. The reasons for these differences are not immediately apparent and need to be explored further. Interestingly, in our analysis, the highest intake level for dairy (quartile 4) for children and adults older than 9 years was well below the dietary recommendations, suggesting that even better B vitamin status could be expected if we compared those who met dairy recommendations with non-consumers or those not meeting the recommendations. Additionally, the associations between milk intake and B vitamin status could be, at least in part, due to the fortified breakfast cereals or other foods that are co-consumed with milk.

Dairy is an important food group and an important component of a healthy diet. Dairy, including fat-free and low-fat (1%) milk, yogurt and cheese are included in the Healthy Dietary Patterns developed by USDA and released as part of the Dietary Guidelines for Americans 2020–2025 [9]. In these patterns, three cups eq dairy per day are included for a 2000 kcal diet [9]. MyPlate also recommends a daily intake of 2–2.5 cups of dairy for children and 3 cups of dairy for adolescents and adults [15]. In our present analysis, mean intake of dairy was less than the recommended levels in all age groups. It is interesting to note that even the highest quartile of dairy intake was less than the recommended levels for adolescents and adults. Earlier analyses presented in the DGA also indicated that only about 20% of adults, 34% of adolescents and 65% of children drink milk as a beverage on a given day and about 90% of the US population does not meet dairy recommendations [9].

There is ongoing discussion and debate about climate and other environmental impacts from animal agriculture (or animal-sourced foods), while ensuring food security for a projected global population of 9–10 billion by 2050. Indeed, some have advocated removing or limiting animal-sourced foods, including dairy, from the diet to minimize environmental impacts [21,22,23,24]. However, such recommendations that account for primarily the environmental impact of foods do not account for how these recommendations may affect nutrient intake. While removing or limiting animal foods from the diet may lead to lower greenhouse gas emissions, trade-offs, such as nutritional inadequacies, may occur as well. This runs counter to public health, especially in vulnerable populations, including those who are pregnant/lactating, in early childhood and elderly, where ensuring adequate nutrition is key. Thus, recommending limiting dairy or other animal-sourced foods could have potential unintended consequences [41,42,43,44].

The strengths of this study include the use of a large nationally representative sample, achieved through combining several sets of NHANES data releases and the use of key covariates to adjust data to remove potential confounding factors; however, even with these covariates, some residual confounders may still exist. A major limitation of this study is the use of a cross-sectional study design, which cannot be used to determine cause and effect. The dietary intake data were self-reported 24 h dietary recalls, relying on memory, and are potentially subject to reporting bias; however, 24 h dietary recall has been one of the most validated tools compared to other dietary assessment tools and provides valuable information with adequate accuracy for intake [45]. Additionally, the results of this study may not actually reflect the effect of dairy products/milk on B vitamin status, since other foods typically consumed with milk (e.g., fortified cereals) were not examined or controlled.

## 5. Conclusions

In conclusion, the results show that intake of total dairy and its individual components’ consumption was associated with improved status and reduced risk of inadequacy of folate, vitamin B_6_ and vitamin B_12_. Dairy products are key sources of macro- and micronutrients; however, their intake is lower than recommended levels. Encouraging dairy consumption may be an effective strategy for improving micronutrient status and achieving a healthier dietary pattern. The current findings provide additional evidence to support dietary recommendations for dairy and dairy products.

## Figures and Tables

**Table 1 nutrients-14-02441-t001:** Mean and quartiles of intake of dairy products among consumers of different age groups, gender-combined data NHANES 2001–2018.

Age (Years)		*n*	Mean ± SE	Quartile 1	Quartile 2	Quartile 3	Quartile 4
2–8	Total dairy (cup eq)	11,478	2.21 ± 0.02	<1.18	1.18 to <1.96	1.96 to <2.96	≥2.96
	Milk (cup eq)	11,112	1.58 ± 0.02	<0.67	0.67 to <1.33	1.33 to <2.18	≥2.18
	Yogurt (cup eq)	1527	0.46 ± 0.01	<0.23	0.23 to <0.40	0.40 to <0.61	≥0.61
	Cheeses (cup eq)	8432	0.78 ± 0.02	<0.25	0.25 to <0.60	0.60 to <1.08	≥1.08
9–18	Total dairy (cup eq)	15,995	2.17 ± 0.03	<0.93	0.93 to <1.80	1.80 to <2.94	≥2.94
	Milk (cup eq)	14,489	1.38 ± 0.02	<0.35	0.35 to <1.05	1.05 to <2.00	≥2.00
	Yogurt (cup eq)	918	0.48 ± 0.02	<0.22	0.22 to <0.43	0.43 to <0.70	≥0.70
	Cheeses (cup eq)	12,170	1.11 ± 0.02	<0.41	0.41 to <0.79	0.79 to <1.48	≥1.48
19–50	Total dairy (cup eq)	22,013	1.83 ± 0.02	<0.66	0.66 to <1.40	1.40 to <2.47	≥2.47
	Milk (cup eq)	18,865	0.98 ± 0.02	<0.14	0.14 to <0.51	0.51 to <1.38	≥1.38
	Yogurt (cup eq)	1861	0.57 ± 0.01	<0.30	0.30 to <0.55	0.55 to <0.77	≥0.77
	Cheeses (cup eq)	16,005	1.20 ± 0.01	<0.44	0.44 to <0.86	0.86 to <1.58	≥1.58
51+	Total dairy (cup eq)	19,650	1.51 ± 0.02	<0.54	0.54 to <1.17	1.17 to <2.09	≥2.09
	Milk (cup eq)	18,027	0.92 ± 0.01	<0.17	0.17 to <0.56	0.56 to <1.27	≥1.27
	Yogurt (cup eq)	1788	0.57 ± 0.01	<0.37	0.37 to <0.56	0.56 to <0.75	≥0.75
	Cheeses (cup eq)	11,278	0.92 ± 0.01	<0.34	0.34 to <0.67	0.67 to <1.22	≥1.22

**Table 2 nutrients-14-02441-t002:** RBC folate status (ng/mL RBC) by dairy intake in different age groups.

	Age (Years)	Non- Consumers	Consumers				Linear Trend	
			Quartile 1	Quartile 2	Quartile 3	Quartile 4	Beta ± SE	*p*
Total dairy (cup eq)	2–8	457 ± 23	438 ± 6	443 ± 7	430 ± 7	427 ± 7	−4.31 ± 2.04	0.0363
	9–18	379 ± 11	387 ± 5	392 ± 5	396 ± 5	399 ± 6	1.63 ± 1.33	0.2204
	19–50	388 ± 9	388 ± 5	405 ± 4	421 ± 5 *	424 ± 5 *	5.02 ± 1.15	<0.0001
	51+	493 ± 11	499 ± 7	528 ± 7 *	547 ± 8 *	540 ± 8 *	9.36 ± 2.07	<0.0001
Milk (cup eq)	2–8	444 ± 11	449 ± 8	440 ± 7	436 ± 8	413 ± 6 *	−12.5 ± 2.4	<0.0001
	9–18	384 ± 7	383 ± 5	407 ± 5 *	395 ± 5	391 ± 6	−1.03 ± 1.88	0.5869
	19–50	395 ± 4	402 ± 4	414 ± 5 *	421 ± 5 *	410 ± 6	−0.42 ± 1.50	0.7775
	51+	524 ± 8	503 ± 7	528 ± 8	548 ± 8	529 ± 8	3.21 ± 2.70	0.2364
Yogurt (cup eq)	2–8	432 ± 5	452 ± 17	449 ± 15	453 ± 15	451 ± 12	27.9 ± 10.9	0.0117
	9–18	390 ± 4	426 ± 23	439 ± 21	433 ± 17	433 ± 19	59.2 ± 16.7	0.0006
	19–50	405 ± 3	447 ± 12 *	449 ± 14 *	428 ± 12	431 ± 11	29.8 ± 8.4	0.0005
	51+	521 ± 5	555 ± 14	580 ± 18 *	605 ± 21 *	552 ± 19	64.8 ± 15.5	<0.0001
Cheeses (cup eq)	2–8	427 ± 6	429 ± 8	425 ± 7	442 ± 8	451 ± 10	14.8 ± 4.9	0.0028
	9–18	392 ± 6	393 ± 7	383 ± 5	394 ± 6	406 ± 6	4.33 ± 2.18	0.0492
	19–50	401 ± 5	388 ± 5 *	408 ± 5	418 ± 5 *	431 ± 5 *	9.45 ± 1.65	<0.0001
	51+	511 ± 7	527 ± 8	523 ± 9	547 ± 9 *	554 ± 9 *	14.5 ± 3.7	0.0002

NHANES 2001–2018, gender-combined data adjusted for age, gender and ethnicity. Data presented as least square means (LSM) ± standard error (SE). Superscript * indicates significant difference from non-consumers at *p* < 0.01. β is the regression coefficient indicating the change in RBC folate status per-cup equivalent of dairy variables, ng/mL change in RBC folate per cup eq.

**Table 3 nutrients-14-02441-t003:** Serum folate status (ng/mL) by dairy intake in different age groups.

	Age (Years)	Non- Consumers	Consumers				Linear Trend	
			Quartile 1	Quartile 2	Quartile 3	Quartile 4	Beta ± SE	*p*
Total dairy (cup eq)	2–8	27.8 ± 1.3	27.1 ± 0.4	27.8 ± 0.4	27.1 ± 0.4	25.7 ± 0.4	−0.56 ± 0.13	<0.0001
	9–18	18.8 ± 0.5	20.4 ± 0.2 *	20.6 ± 0.2 *	21.6 ± 0.3 *	21.9 ± 0.3 *	0.31 ± 0.07	0.0001
	19–50	16.0 ± 0.9	15.4 ± 0.2	17.0 ± 0.3	17.3 ± 0.2	17.5 ± 0.2	0.31 ± 0.09	0.0011
	51+	21.8 ± 0.8	22.5 ± 0.4	23.6 ± 0.3	24.7 ± 0.4 *	24.0 ± 0.3 *	0.4 ± 0.12	0.0010
Milk (cup eq)	2–8	26.5 ± 0.9	27.2 ± 0.4	27.5 ± 0.4	27.3 ± 0.5	25.8 ± 0.4	−0.61 ± 0.17	0.0004
	9–18	19.3 ± 0.3	20.0 ± 0.2	21.2 ± 0.3 *	21.6 ± 0.3 *	22.2 ± 0.3 *	0.53 ± 0.10	<0.0001
	19–50	15.4 ± 0.3	15.7 ± 0.2	16.8 ± 0.3 *	18.1 ± 0.2 *	17.7 ± 0.3 *	0.54 ± 0.17	0.0022
	51+	22.3 ± 0.5	21.7 ± 0.4	23.4 ± 0.4	24.8 ± 0.3 *	25.1 ± 0.4 *	0.85 ± 0.17	0.0000
Yogurt (cup eq)	2–8	26.9 ± 0.3	28.4 ± 1.0	27.1 ± 1.1	27.4 ± 1.0	26.1 ± 1.0	−0.47 ± 0.80	0.5589
	9–18	20.9 ± 0.2	21.4 ± 1.0	23.0 ± 1.0	22.4 ± 1.0	23.3 ± 0.9	2.91 ± 0.85	0.0008
	19–50	16.5 ± 0.1	18.2 ± 0.9	18.6 ± 0.6 *	19.1 ± 0.7 *	21.0 ± 2.5	3.68 ± 1.31	0.0056
	51+	23.5 ± 0.2	23.8 ± 0.7	26.6 ± 1.2 *	25.4 ± 1.4	23.4 ± 1.0	1.12 ± 0.86	0.1954
Cheeses (cup eq)	2–8	27.2 ± 0.4	27.2 ± 0.5	27.4 ± 0.6	26.4 ± 0.4	26.3 ± 0.4	−0.44 ± 0.27	0.1008
	9–18	21.0 ± 0.3	21.3 ± 0.3	20.9 ± 0.3	21.0 ± 0.3	21.0 ± 0.3	−0.06 ± 0.10	0.5782
	19–50	16.9 ± 0.3	16.5 ± 0.3	17.0 ± 0.3	16.9 ± 0.2	16.4 ± 0.2	−0.12 ± 0.11	0.2683
	51+	23.4 ± 0.3	25.2 ± 0.4 *	23.7 ± 0.4	23.0 ± 0.4	23.0 ± 0.4	−0.47 ± 0.18	0.0091

NHANES 2001–2018 gender-combined data adjusted for age, gender and ethnicity. Data presented as least square means (LSM) ± standard error (SE). Superscript * indicates significant difference from non-consumers at *p* < 0.01. β is the regression coefficient indicating the change in serum folate status per-cup equivalent of dairy variables, ng/mL serum folate per cup eq.

**Table 4 nutrients-14-02441-t004:** Serum pyridoxal 5′- phosphate (Vitamin B_6_) status (nmol/L) by dairy intake in different age groups.

	Age (Years)	Non- Consumers	Consumers				Linear Trend	
			Quartile 1	Quartile 2	Quartile 3	Quartile 4	Beta ± SE	*p*
Total dairy (cup eq)	2–8	62.1 ± 6.9	63.6 ± 1.7	71.7 ± 2.1	72.9 ± 3.3	75.7 ± 3.6	2.17 ± 0.89	0.0178
	9–18	54.0 ± 4.5	58.7 ± 1.6	56.9 ± 1.3	60.9 ± 1.7	67.3 ± 2.2	1.94 ± 0.56	0.0009
	19–50	53.5 ± 2.9	61.5 ± 2.5	70.6 ± 2.4 *	72.6 ± 2.8 *	71.9 ± 3.1 *	0.70 ± 0.48	0.1506
	51+	64.9 ± 5.0	73.5 ± 3.8	78.8 ± 3.5	85.8 ± 3.7 *	90.0 ± 2.8 *	4.75 ± 1.47	0.0020
Milk (cup eq)	2–8	65.9 ± 6.9	62.6 ± 2.2	72.6 ± 3.1	74.2 ± 2.7	74.3 ± 3.2	2.85 ± 0.97	0.0045
	9–18	55.1 ± 2.4	57.4 ± 1.8	58.7 ± 1.5	63.0 ± 2.2	66.3 ± 2.0 *	2.96 ± 0.81	0.0005
	19–50	64.4 ± 3.7	64.8 ± 2.5	65.5 ± 2.6	74.3 ± 3.1	71.4 ± 3.0	0.92 ± 0.69	0.1851
	51+	77.0 ± 3.6	74.3 ± 4.8	74.9 ± 3.0	85.5 ± 3.3	90.5 ± 4.0 *	5.68 ± 2.03	0.0067
Yogurt (cup eq)	2–8	70.1 ± 1.9	68.4 ± 5.1	107 ± 16	69.7 ± 5.1	69.4 ± 5.8	6.62 ± 6.44	0.3084
	9–18	59.9 ± 1.1	68.1 ± 3.8	71.5 ± 8.6	77.3 ± 9.8	85.0 ± 11.8	25.4 ± 9.9	0.0123
	19–50	66.6 ± 1.8	80.9 ± 8.1	87.8 ± 11	96.0 ± 10.7 *	93.9 ± 10.1 *	30.4 ± 8.1	0.0004
	51+	78.8 ± 2.1	90.5 ± 6.0	104 ± 9.8	109 ± 12	114 ± 10 *	44.5 ± 8.7	<0.0001
Cheeses (cup eq)	2–8	69.6 ± 2.1	72.0 ± 2.6	68.0 ± 2.9	74.3 ± 2.9	72.1 ± 4.9	0.15 ± 1.88	0.9367
	9–18	62.5 ± 1.5	58.0 ± 1.5	60.9 ± 2.3	60.3 ± 2.3	62.1 ± 2.2	0.30 ± 0.58	0.6026
	19–50	66.4 ± 2.4	67.5 ± 3.0	72.8 ± 3.6	68.8 ± 3.0	68.2 ± 3.0	−0.93 ± 0.76	0.2251
	51+	79.5 ± 3.7	90.4 ± 4.2	76.7 ± 3.8	79.8 ± 3.5	82.3 ± 3.7	0.42 ± 2.24	0.8523

NHANES 2003–2010, gender-combined data adjusted for age, gender and ethnicity. Data presented as least square means (LSM) ± standard error (SE). Superscript * indicates significant difference from non-consumers at *p* < 0.01. β is the regression coefficient indicating the change in serum pyridoxal 5′-phosphate status per-cup equivalent of dairy variables, nmol/L of pyridoxal 5′-phosphate per cup eq.

**Table 5 nutrients-14-02441-t005:** Serum vitamin B_12_ status (pg/mL) by dairy intake in different age groups.

	Age (Years)	Non-Consumers	Consumers				Linear Trend	
			Quartile 1	Quartile 2	Quartile 3	Quartile 4	Beta ± SE	*p*
Total dairy (cup eq)	2–8	751 ± 60	779 ± 16	832 ± 16	826 ± 18	917 ± 18	31.1 ± 4.8	<0.0001
	9–18	518 ± 20	551 ± 9	580 ± 10 *	616 ± 10 *	650 ± 8 *	18.6 ± 2.4	<0.0001
	19–50	529 ± 41	526 ± 22	513 ± 7	547 ± 9	571 ± 11	12.3 ± 3.4	0.0005
	51+	699 ± 98	555 ± 10	630 ± 23	639 ± 18	620 ± 11	10.3 ± 4.8	0.0371
Milk (cup eq)	2–8	757 ± 31	751 ± 17	830 ± 17	843 ± 19	917 ± 15 *	43.5 ± 6.0	<0.0001
	9–18	541 ± 10	543 ± 8	590 ± 10 *	613 ± 9 *	660 ± 10 *	26.7 ± 3.0	<0.0001
	19–50	517 ± 15	502 ± 12	535 ± 22	552 ± 8	582 ± 13 *	20.7 ± 4.1	<0.0001
	51+	670 ± 51	573 ± 12	575 ± 11	634 ± 27	645 ± 13	15.6 ± 5.6	0.0067
Yogurt (cup eq)	2–8	840 ± 10	865 ± 70	872 ± 43	810 ± 62	899 ± 41	40.7 ± 43.1	0.3496
	9–18	599 ± 6	605 ± 36	650 ± 49	609 ± 37	561 ± 25	−20.5 ± 30.3	0.5033
	19–50	537 ± 7	531 ± 21	592 ± 23	560 ± 20	555 ± 29	28.4 ± 20.5	0.1689
	51+	607 ± 11	657 ± 30	704 ± 54	728 ± 59	655 ± 35	94.1 ± 36.9	0.0129
Cheeses (cup eq)	2–8	849 ± 17	817 ± 17	883 ± 20	818 ± 25	829 ± 21	−16.1 ± 11.8	0.1794
	9–18	591 ± 9	601 ± 10	602 ± 10	596 ± 9	607 ± 12	3.14 ± 4.76	0.5134
	19–50	556 ± 13	536 ± 24	518 ± 8	529 ± 8	546 ± 15	−3.46 ± 3.83	0.3696
	51+	618 ± 17	653 ± 34	584 ± 13	596 ± 18	614 ± 25	−9.96 ± 9.78	0.3117

NHANES 2001–2006 and 2011–2014, gender-combined data adjusted for age, gender and ethnicity. Data presented as least square means (LSM) ± standard error (SE). Superscript * indicates significant difference from non-consumers at *p* < 0.01. β is the regression coefficient indicating the change in serum vitamin B_12_ status per-cup equivalent of dairy variables, pg/mL vitamin B_12_ per cup eq.

**Table 6 nutrients-14-02441-t006:** Linear trends in odds ratios (OR), 99% upper and lower confidence limits (UCL, LCL) for vitamins deficiency/inadequacy across consumer quartiles of intake by age group. Gender-combined data.

Age (Years)	Total Dairy (cup eq)	Milk (cup eq)	Yogurt (cup eq)	Cheese (cup eq)
Inadequate RBC (Erythrocyte) Folate (<140 ng/mL)				
2–8	0.84 (0.46, 1.53)	0.80 (0.48, 1.35)	0.0003 (0.0001, 0.0007) *	0.86 (0.52, 1.43)
9–18	0.73 (0.58, 0.92) *	0.79 (0.64, 0.97) *	0.68 (0.38, 1.23)	0.87 (0.73, 1.04)
19–50	0.83 (0.70, 0.97) *	0.97 (0.83, 1.12)	0.64 (0.36, 1.11)	0.85 (0.77, 0.94) *
51+	0.73 (0.57, 0.92) *	0.78 (0.64, 0.94) *	0.99 (0.61, 1.62)	0.79 (0.63, 0.99) *
Inadequate Serum Folate (<3 ng/mL)				
2–8				
9–18	0.43 (0.27, 0.68) *	0.54 (0.38, 0.78) *	0.82 (0.31, 2.18)	0.63 (0.44, 0.89) *
19–50	0.78 (0.64, 0.94) *	0.88 (0.73, 1.06)	0.57 (0.36, 0.90) *	0.86 (0.74, 1.01)
51+	0.69 (0.49, 0.96) *	0.74 (0.59, 0.93) *	0.71 (0.33, 1.53)	0.84 (0.65, 1.08)
Inadequate Serum Vitamin B_6_ (<20 nmol/L)				
2–8	0.96 (0.71, 1.29)	1.02 (0.80, 1.30)	0.67 (0.43, 1.05)	1.10 (0.84, 1.45)
9–18	0.95 (0.81, 1.12)	0.94 (0.79, 1.11)	0.89 (0.72, 1.10)	1.03 (0.92, 1.15)
19–50	0.89 (0.80, 0.98) *	0.91 (0.85, 0.97) *	0.77 (0.60, 0.99) *	0.95 (0.88, 1.03)
51+	0.84 (0.78, 0.91) *	0.86 (0.80, 0.92) *	0.78 (0.65, 0.94) *	0.95 (0.88, 1.04)
Serum Vitamin B_12_ Deficient (<200 pg/mL)				
2–8				
9–18	0.49 (0.27, 0.88) *	0.60 (0.45, 0.81) *	1.08 (0.45, 2.58)	0.76 (0.46, 1.25)
19–50	0.79 (0.59, 1.06)	0.81 (0.64, 1.03)	1.05 (0.67, 1.65)	0.88 (0.72, 1.07)
51+	0.75 (0.63, 0.89) *	0.77 (0.66, 0.88) *	0.77 (0.58, 1.01)	0.98 (0.83, 1.16)

NHANES 2001–2018 data for serum and RBC folate; NHANES 2003–2010 data for serum vitamin B_6_; NHANES 2001–2006 and 2011–2014 data for serum vitamin B_12_ inadequacy/deficiency. Superscript * indicates significant linear trend across consumer quartiles of intake at *p* < 0.01. Inadequate serum folate and serum vitamin B_12_ deficient results for age 2–8 years are not indicated as there were less than 5 subjects per quartile.

## Data Availability

All data obtained for this study are publicly available at: http://www.cdc.gov/nchs/nhanes/ (accessed on 26 April 2021).

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
