# Peer review of "Association between Intake of Total Dairy and Individual Dairy Foods and Markers of Folate, Vitamin B6 and Vitamin B12 Status in the U.S. Population"

_nutrients, 2022, doi:10.3390/nu14122441_

Round 1
Reviewer 1 Report
Using data from NAHNES, Cifelli et al. analyzed the intake of dairy products in relation RBC folate and the serum concentrations of folate, vitamin B6 and B12. The results show that an increasing consumption of dairy products is associated with an improved status of folate, B6 and B12. In addition, adolescent and adults often do not meet the recommended daily intake. In the context of changing nutritional habits, this study is scientifically important as is provides robust support for exisiting recommendations to consume dairy products.
I have only a very few questions and suggestions that might be considered by the authors:
11) Are results for function markers of folate, B6 and B12 available (e.g. homocysteine, MMA, holo-transcobalamin). If yes, these results should be included.
22) Line 31: In addition to the relation between homocysteine and coronary heart disease, the authors should also mention stroke. In addition, the authors should quote a recent review paper by Herrmann M and Herrmann W, Nutrients. 2022 Mar 28;14(7):1412. doi: 10.3390/nu14071412.
33) The analytical methods for all biomarkers should at least be mentioned briefly.
44) How do the authors explain the opposite direction of ß-values for RBC and serum folate?
Author Response
Reviewer 1:
Using data from NAHNES, Cifelli et al. analyzed the intake of dairy products in relation RBC folate and the serum concentrations of folate, vitamin B6 and B12. The results show that an increasing consumption of dairy products is associated with an improved status of folate, B6 and B12. In addition, adolescent and adults often do not meet the recommended daily intake. In the context of changing nutritional habits, this study is scientifically important as is provides robust support for exisiting recommendations to consume dairy products.
I have only a very few questions and suggestions that might be considered by the authors:
- Are results for function markers of folate, B6 and B12 available (e.g. homocysteine, MMA, holo-transcobalamin). If yes, these results should be included.
Authors Response: While there are data available for homocysteine and methyl malonic acid, unfortunately these data are only available for four cycles of NHANES that we used. That said, we thank the reviewer for raising this point as this might be an interesting follow-up research opportunity (albeit with more limited data than in the current manuscript).
- Line 31: In addition to the relation between homocysteine and coronary heart disease, the authors should also mention stroke. In addition, the authors should quote a recent review paper by Herrmann M and Herrmann W, Nutrients. 2022 Mar 28;14(7):1412. doi: 10.3390/nu14071412.
Authors Response: Added “stroke” in line 31 and included the reference (#4) as suggested.
- The analytical methods for all biomarkers should at least be mentioned briefly.
Authors Response: Brief analytical methods for all biomarkers are included in line 113-115.
- How do the authors explain the opposite direction of ß-values for RBC and serum folate?
Authors Response: Discussion on the opposite direction for folate is included in line 251-253.
Reviewer 2 Report
This manuscript deals with an interesting topic, with the possibility of being applied in clinical practice. As a Suggestion, i propose not only to compare in age groups, but also to compare betiene menos and women population.
It would be interesting to see if there is any European study to compare the results obtained in American population.
Author Response
This manuscript deals with an interesting topic, with the possibility of being applied in clinical practice. As a Suggestion, i propose not only to compare in age groups, but also to compare betiene menos and women population.
Authors Response: We thank the reviewer for raising this point. While our regression analyses indicated there were some gender differences we felt this was most likley to higher calories and thus nutrient intake of males as compared to females. And thus to properly assess differences in males and females we would need furhter analyses adjusting for calories/nutrient intake and at this time we felt it was beyond the scope of the current manuscript, but certainly will make an excellent follow-up study.
It would be interesting to see if there is any European study to compare the results obtained in American population.
Authors Response: Results of studies on diverse populations (European and also Asian) are discussed in lines 254-274.